# The Role of Electron Transfer in the Fragmentation of Phenyl and Cyclohexyl Boronic Acids

**DOI:** 10.3390/ijms20225578

**Published:** 2019-11-08

**Authors:** Ana Isabel Lozano, Beatriz Pamplona, Tymon Kilich, Marta Łabuda, Mónica Mendes, João Pereira-da-Silva, Gustavo García, Pedro M. P. Gois, Filipe Ferreira da Silva, Paulo Limão-Vieira

**Affiliations:** 1Atomic and Molecular Collisions Laboratory, CEFITEC, Department of Physics, Universidade NOVA de Lisboa, 2829-516 Caparica, Portugal; ai.lozano@fct.unl.pt (A.I.L.); b.pamplona@campus.fct.unl.pt (B.P.); mf.mendes@fct.unl.pt (M.M.); jvp.silva@campus.fct.unl.pt (J.P.-d.-S.); f.ferreiradasilva@fct.unl.pt (F.F.d.S.); 2Department of Theoretical Physics and Quantum Information, Gdańsk University of Technology, Narutowicza 11/12, 80-233 Gdańsk, Poland; tymon.kilich@pg.edu.pl (T.K.); marta.labuda@pg.edu.pl (M.Ł.); 3Instituto de Física Fundamental, Consejo Superior de Investigaciones Científicas (CSIC), Serrano 113-bis, 28006 Madrid, Spain; g.garcia@csic.es; 4Institute for Medicines and Pharmaceutical Sciences (iMed.ULisboa), Faculty of Pharmacy, Universidade de Lisboa, Av. Prof. Gama Pinto, 1649-003 Lisbon, Portugal; pedrogois@ff.ul.pt

**Keywords:** boronic acids, electron transfer, TOF mass spectrometry, negative ion formation

## Abstract

In this study, novel measurements of negative ion formation in neutral potassium-neutral boronic acid collisions are reported in electron transfer experiments. The fragmentation pattern of phenylboronic acid is comprehensively investigated for a wide range of collision energies, i.e., from 10 to 1000 eV in the laboratory frame, allowing some of the most relevant dissociation channels to be probed. These studies were performed in a crossed molecular beam set up using a potassium atom as an electron donor. The negative ions formed in the collision region were mass analysed with a reflectron time-of-flight mass spectrometer. In the unimolecular decomposition of the temporary negative ion, the two most relevant yields were assigned to BO^−^ and BO_2_^−^. Moreover, the collision-induced reaction was shown to be selective, i.e., at energies below 100 eV, it mostly formed BO^−^, while at energies above 100 eV, it mostly formed BO_2_^−^. In order to further our knowledge on the complex internal reaction mechanisms underlying the influence of the hybridization state of the boron atom, cyclohexylboronic acid was also investigated in the same collision energy range, where the main dissociation channel yielded BO_2_^−^. The experimental results for phenyl boronic acid are supported by ab initio theoretical calculations of the lowest unoccupied molecular orbitals (LUMOs) accessed in the collision process.

## 1. Introduction

During the last decades, the international scientific community has been paying considerable attention to understanding the role of different molecules used as drug-precursors within physiological environments. Nevertheless, some physico-chemical properties and action mechanisms at the molecular level remain unknown. Therefore, a deeper knowledge of such processes may help our understanding of their molecular reactivity and thereby improve the clinical utility of these molecules. In this study, we focused on two different boronic acid-containing compounds: the phenyl boronic acid C_6_H_5_B(OH)_2_ (PBA) and the cyclohexyl boronic acid C_6_H_11_B(OH)_2_ (CHBA). Boronic compounds are extensively used in pharmaceutical drug design [1] due to the similarity between the electronic structure of the boron and carbon atoms. In particular, in the case of boronic acids, we can highlight different applications, such as fluorescent cellular dyes [2], saccharide sensors [3], dopamine sensors, and several enzymatic inhibitors [4]. Additionally, their use in HIV treatment, cancer therapy, diabetes, and obesity has been proven to be efficient [5].

Moreover, study on the interactions between low-energy electrons (LEE) with biologically relevant molecules has been extensively described since pioneering experiments performed by Sanche and co-workers revealed that those electrons may play an important role in inducing damage in biological systems, especially in DNA and RNA [6,7]. Nowadays, it is well-established that LEE can alter the internal state of a molecule via resonant attachment, which may lead to molecular dissociation (dissociative electron attachment, DEA). In such a process, a temporary negative ion (TNI) is formed and is highly likely to dissociate into stable fragment anions and neutral radical species [8], as long as it successfully competes with autodetachment. Although free LEE may be found in physiological environments as a secondary product from the interaction of radiation with biological matter (e.g., through photoelectron processes and/or Compton scattering), the role of electron transfer processes seems to be more prevalent, under physiological conditions, in comparison with free electron attachment processes [9]. Another interesting aspect of electron transfer processes is the relevance of a Coulomb complex formed in the vicinity of the electron donor projectile and the electron acceptor target, which may significantly change the fragmentation pathways not accessible in free electron capture [10,11,12]. Since boronic acids have been expanding their influence as biologically useful compounds, even in drugs used in cancer treatment, it is relevant to recognize which fragments are produced either via DEA or electron transfer processes, in order to obtain a deeper knowledge on their behavior and fundamental properties and thereby improve their clinical utility.

The aim of this study is to characterize the dissociation pattern of the above mentioned boronic acids upon electron transfer in atom-molecule collision experiments in the energy range of 10–1000 eV, paying special attention to the differences found from the fragmentation yields due to the different electronic structures of the cyclic ring (see Figure 1). In these experiments, an electron donor provided by a neutral potassium atom K was delivered to the neutral target molecule yielding negative ions that were time-of-flight mass analysed. The significance of these experiments has been extensively proven with other molecules of high biological relevance, such as DNA/RNA bases thymine and uracil [13], as well as the prototype of DNA/RNA pyrimidine [12], showing that hydrogen loss is made site- and bond-selective as a function of the collision energy. In addition, different molecules used as drug-precursors as radiosensitizers have also been studied [10,11].

## 2. Results and Discussion

In this section, we discuss the anionic fragmentation pattern of PBA upon neutral potassium atom collisions. Dissociative electron transfer TOF mass spectra were recorded at different lab-frame collision energies (10–1000 eV and 6.8–681.5 eV in the centre–of–mass frame and from now on, referred to as available energy), and a complete analysis of the anionic species formed during the collisions was performed. The mass spectra were normalized to the acquisition time and the neutral potassium current, yet in no absolute values. Further comparison with CHBA was also investigated and a thorough discussion will be presented below. As far as the authors are aware, these are the first sets of electron transfer studies with these molecules, and there no other datasets available in the literature to compare them with.

In Table 1, all negative ions assigned for both PBA and CHBA molecules for the different probed collision energies are listed. Figure 2 shows the TOF mass spectra of PBA at 70 eV (A) and 300 eV (B) lab-frame collision energies (43 and 200 eV available energies), where the fragmentation increases with increasing collision energy. Such behavior is expected due to the excess energy that is available in the system leading to further fragmentation. The lowest energy recorded in the present experiments was 6.8 eV (2.5 eV available energy). However, only at 13 eV was negative ion formation from PBA observed. This can be rationalized in terms of the molecular orbitals involved in the electron transfer process. Figure 8 shows that only an electron promotion to LUMO+2 (12.1 eV), which is localized along the phenyl ring with a π* character and C1-B bond contribution, may result in fragmentation. Note that accessing the LUMO (4.8 eV) and LUMO+1 (6.0 eV), both with a π* character along the ring, may result in strong autodetachment yielding no fragment anion. Such an assumption seems reasonable since, at low energies, the estimated collision time is of the order of 200 fs. The TOF mass spectra of PBA are essentially dominated by BO^−^ (27 *m*/*z*) and BO_2_^−^ (43 *m*/*z*), and no parent anion was detected. Other relevant but less intense fragments were also assigned to C_2_H^−^ (25 *m*/*z*), O_2_^−^ (32 *m*/*z*), C_2_BO^−^/C_3_H_4_B^−^ (51 *m*/*z*), C_5_H_5_^−^ (65 *m*/*z*), and C_5_H_4_BO^−^ (91 *m*/*z*). Note that isobaric contributions at 51 *m*/*z* cannot be distinguished unambiguously due to the mass resolution ≈ 700. The TOF mass spectra also show that there are multiple intense peaks separated by one atomic mass, so the ratio between the total yield of these anions was plotted and is depicted in Figure 3. This is a very relevant aspect that leads to an increase in the number of fragment anions detected.

Boron’s natural isotopic abundances are ^10^B (10 *m*/*z*) and ^11^B (11 *m*/*z*), and the former ≈ ¼ of the latter [14]. Therefore, if this ratio is kept constant as a function of the collision energy, we can assume that any two peaks belong to the same fragment containing the two different boron isotopes. A close inspection of Figure 3, within the associated uncertainty, reveals that such a ratio is, roughly speaking, kept constant for fragment anions *m*/*z* 42/43, 50/51, and 90/91, which may be assigned to ^10^BO_2_^−^/^11^BO_2_^−^, C_2_^10^BO^−^(C_3_H_4_^10^B^−^)/C_2_^11^BO^−^(C_3_H_4_^11^B^−^), and C_5_H_4_^10^BO^−^/C_5_H_4_^11^BO^−^, respectively. Note that such a ratio is consistently much higher than 25% for *m*/*z* 64 and *m*/*z* 65. Hence, the heavier fragment (*m*/*z* 65) is unlikely to contain boron and was assigned to C_5_H_5_^−^. Another interesting aspect of the anion ratio in Figure 3 pertains to the case of *m*/*z* 26 and *m*/*z* 27 mostly being above 100 eV available energy, which do not remain constant, but increase with increasing available energy. This can be related to the opening of another dissociation channel, apart from the isobaric contribution of ^10^BO^−^ and ^11^BO^−^, which is assigned to C_2_H_2_^−^ with an enthalpy of formation (∆*H*_0_) of 12.33 eV. This value was obtained from the sum of the bond dissociation energies of two C–C (Table 2) minus the electron affinity of C_2_H_2_ (Table 3).

### 2.1. BO^−^ and BO_2_^−^

The negative ions BO^−^ and BO_2_^−^ are the most intense fragments observed in the TOF mass spectra of PBA at all collision energies and account for 40% of the total anion yield. This can be expected considering the high electron affinities of their neutrals (2.51 and 4.46 eV) [15]. Figure 4 shows BO^−^ and BO_2_^−^ branching ratios as a function of the available energy for PBA. At lower energies (below 100 eV), the fragmentation pattern is dominated by the boron monoxide anion (BO^−^), which becomes the only discernible fragment in the TOF mass spectrum at 12.8 eV available energy. Recalling that at 6.8 eV, no traces of any negative ions formed, the appearance of the energy of BO^−^ lies between 6.8 and 12.8 eV. Figure 8 shows that the LUMO+3 at 14.4 eV has a π* character, while LUMO+4 at 16.7 eV has a σ* character. Therefore, the initial access, mainly from the HOMO to the π* state and subsequent intramolecular electron transfer into the highly antibonding σ* state, may enhance an effective C–B breaking pathway. This is achieved in electron transfer studies since the presence of the K^+^ ion in the vicinity of the TNI may suppress autodetachment long enough for successful competition of the fragmentation pathway. However, efficient bond breaking should proceed through the access of σ* states and achieving LUMO+4 may lend support to such an assumption, meaning that a high ion yield is obtained at a low collision energy. Therefore, we expect that the BO^−^ threshold may lie closer to 12.8 eV.

Boron dioxide (BO_2_^−^) is detected at an available energy of 16.3 eV and is the most predominant fragment ion for higher energies (above 100 eV), showing the opposite energy dependence to BO^−^ (Figure 4). This behavior shows how selective the collision-induced dissociation process is, which is simply achieved by tuning the proper available energy. The BO^−^ branching ratio in Figure 4 shows that the dissociation of PBA by electron transfer is dependent on the collision energy and subsequently on the collision time, whereas for BO_2_^−^, its yield shows no significant energy dependence above 200 eV. The access through LUMO+4 and/or LUMO+5 at 16.7 and 18.8 eV of the σ* character may be strongly related to the main character of the dissociation mechanism. Notwithstanding, since the electron affinity of BO_2_ is significantly higher than BO (see Table 2), energy constraints cannot solely explain site selectivity, where the electronic structure of the associated transient precursor ions accessed by electrons of different energies (either shape or core excited resonances) has been suggested as the main effect responsible for such an achievement [13].

In Figure 4A, dashed markers schematically represent the necessary bonds to break to yield BO^−^. Hence, such anion formation in electron transfer can proceed through the following reactions:(1)$$K^{0}+C_{6}H_{5}B{\left(OH\right)}_{2}\to \left[K^{+}+{\left(C_{6}H_{5}B{\left(OH\right)}_{2}{}^{-}\right)}^{\#}\right]\to K^{+}+BO^{-}+C_{6}H_{5}^{•}+OH^{•}+H^{•}$$
(2)$$K^{0}+C_{6}H_{5}B{\left(OH\right)}_{2}\to \left[K^{+}+{\left(C_{6}H_{5}B{\left(OH\right)}_{2}{}^{-}\right)}^{*}\right]\to K^{+}+BO^{-}+C_{6}H_{5}^{•}+H_{2}O,$$
where # refers to a temporary negative ion (TNI) formed with an excess of internal energy. The reaction enthalpy (∆*H*_0_) of (1) is given by the sum of the bond dissociation energies of C–B, B–O, and O–H, minus the electron affinity (EA) of BO. Therefore, a value of 14.93 eV is obtained considering the electron affinity of BO in Table 2 and the bond dissociation energies from Table 3.

The possible bond excisions involved in the formation of BO_2_^−^ are also schematically represented in Figure 4B, where a calculated ∆*H*_0_ value for reaction (3) is 9.06 eV, while for reaction (4), it is 3.91 eV.
(3)$$K^{0}+C_{6}H_{5}B{\left(OH\right)}_{2}\to \left[K^{+}+{\left(C_{6}H_{5}B{\left(OH\right)}_{2}{}^{-}\right)}^{\#}\right]\to K^{+}+BO_{2}^{-}+C_{6}H_{5}^{•}+H_{2}$$
(4)$$K^{0}+C_{6}H_{5}B{\left(OH\right)}_{2}\to \left[K^{+}+{\left(C_{6}H_{5}B{\left(OH\right)}_{2}{}^{-}\right)}^{\#}\right]\to K^{+}+BO_{2}^{-}+C_{6}H_{5}^{•}+H^{•}+H^{•}$$

Reactions (1), (2) and (3), (4) may lead to the formation of radical species, OH^●^ and H^●^, which are known to be relevant within the context of indirect damage in the biological environment. These highly reactive species have been shown to be associated with an increase of DNA strand-breaks [16,17] and molecular oxidation. The formation of BO^−^ can yield a water molecule (reaction (1)) and the reaction threshold is obtained at 10.41 eV.

Note that in (3) and (4), the loss of two H atoms may proceed through reactions yielding H + H and/or H_2_. Unfortunately, we are not aware of any dissociative electron attachment studies from which to obtain information on the energy position of the resonances involved in these processes. However, we may recall recent electron transfer studies on adenine and selected analogues, where the loss of H_2_ was attributed to 0.7 eV resonance, whereas H + H formation was accessible from two high energy resonances at 7 and 10.6 eV [18]. There is a considerable difference between the minimum energy required for BO^−^ (~15 eV) and BO_2_^−^ (~9 eV) formation. As such, BO_2_^−^ should be the fragment ion with the lowest experimental threshold energy. However, this is not observed in the present results and if a similar H + H and/or H_2_ process in reaction (3) and (4) holds, this will lower its threshold of formation.

In atom–molecule collisions, electron transfer occurs from the projectile atom (K) to the target molecule (M), resulting in ion-pair formation. In the collision complex, (K^+^ M^−^^#^), the interaction is mainly coulombic in nature and may determine the sort of anionic exit channel. The presence of a potassium cation will be relevant for larger collision times, meaning a longer transit time in the vicinity of the TNI. This may either result in autodetachment competing with dissociation or allow an intramolecular energy distribution through the different degrees of freedom, enhancing a particular dissociation channel.

PBA is a planar molecule with a very particular electronic structure, where the carbon atoms in the phenyl ring are *sp*^2^-hybridized, which leads to six delocalized electrons over the ring. The boron atom is bonded with *sp*^2^ hybridization that also provides a planar geometry. However, it has a vacant *p* orbital out of the plane of the molecule with the same spatial orientation as the delocalized π electrons of the phenyl ring. Such an arrangement confers a particular chemical environment, where intramolecular electron transfer processes may occur, dictating the sort of dissociation pattern upon the electron transfer to PBA.

In order to confirm this rationale, we have investigated electron transfer in a cyclohexylboronic acid (CHBA) molecule, since it has no π electrons (Figure 1), i.e., the carbon atoms are *sp*^3^-hybridized.

A close inspection of the negative ion fragmentation patterns from the TOF mass spectra of CHBA and PBA at 100 and 200 eV collision energies, in Figure 5 and Figure 6, show significant differences. The first noticeable difference is that the fragmentation pattern of PBA is much richer than in CHBA. Another interesting aspect of the PBA fragmentation pattern pertains to the ion yields obtained at lower collision energies (not shown here), which mainly result in BO^−^ and BO_2_^−^ formation with unbearably discernible evidence of other fragments stemming from phenyl ring opening. In K + PBA collisions, the electron may be transferred either to the phenyl ring or the boron-containing end. Given that EA(C_6_H_5_^•^) = (1.0960 ± 0.0060) eV and the electron affinity of B(OH)_2_ is unknown but certainly lower than EA(BO_2_) = (4.460 ± 0.030) eV (see Table 2), in the low-collision energy regime (typically < 100 eV) with longer collision times, initial electron capture to the phenyl ring may be transferred to the boron site as long as the nuclear wavepacket survives long enough for efficient diabatic crossing of the π* and σ* states. This is an assertion of the efficient intramolecular charge transfer since the collision complex formed by the potassium cation and the TNI may enhance such a process, preferentially yielding BO^−^ and BO_2_^−^. In the case of CHBA, BO_2_^−^ and BO^−^ formation are the major product anions of the fragmentation yield for all collision energies, where the former amounts to more than half of the total number of fragment anions detected. The formation of boron-containing anions seems plausible, because the electron transfer process may mainly occur at the –B(OH)_2_ end given that EA(C_6_H_11_^•^) = (−0.24 ± 0.11) eV (Table 2). The loss of hydrogen atoms from the –B(OH)_2_ end seems plausible from the inspection of the MOs in Figure 8, in both π* and σ* antibonding orbitals, due to the favorable electron spin density over the phenyl ring carbon atoms and boron atom. However, these MOs may be easily accessible within the range of collision energies probed in these experiments. Another relevant aspect of the TOF mass spectra yields is related to the relative intensities of BO_2_^−^ and BO^−^ in PBA and CHBA. While, in the former (PBA), BO^−^ prevails for lower energies, meaning that LUMO+4 may be mainly accessed, for higher collision energies, LUMO+5 yielding BO_2_^−^ becomes relevant. The spin densities of such MOs show that the σ* antibonding character over the phenyl ring involves more carbon atoms in LUMO+5 than in LUMO+4, meaning that fragment anions stemming from the ring and containing boron atoms are also enhanced. As far as the latter molecule is concerned (CHBA), although we have no information about the MOs, from Figure 8, we may anticipate that similar spin densities of the –B(OH)_2_ end may be expected. Since the π*/σ* coupling is no longer relevant due to the absence of π ring MOs, the antibonding character of the B–C bond remains, preferentially yielding BO_2_^−^. For the collision energies probed, the available energy in the centre-of-mass frame is well above the reaction threshold, resulting in the loss of hydrogen atoms from –B(OH)_2_.

### 2.2. Other Relevant Fragments

In the case of PBA, apart from boron monoxide and boron dioxide anions, other boron-containing fragments have been assigned. Among these, three different isotopic fragments corresponding to *m*/*z* 91/90, 64/63, and 51/50 (see Table 1) have been identified as the most intense fragments in comparison with other boron-containing anions. The fragments *m*/*z* 50/51 and 90/91 are formed for all energies above 23 eV, while *m*/*z* 63/64 has a threshold above 43 eV. The electron spin densities in Figure 8 show that the initial access is mainly from HOMO-1 and HOMO-2 to the MOs yielding BO^−^ and/or BO_2_^−^, and the antibonding MOs also reveal a substantial phenyl ring weakness of the σ* character (LUMO+4 and LUMO+5). A careful inspection of the TOF mass spectrum in Figure 2A, shows that anions containing boron and part of the phenyl ring carbon atoms (e.g., C_2_BO^−^) are also formed, but with lower yields. Another interesting aspect is the fact that the formation of these fragments implies ring opening. Here, it is important to highlight that among all ions formed, just the heavier fragment (with a low intensity), C_6_H_4_BO^−^
*m*/*z* 103, does not result from the ring opening.

Regarding fragment anions that do not contain boron, four with considerable intensities have been assigned to C_2_H^−^ (*m*/*z* 25), C_2_H_2_^−^ (*m*/*z* 26), O_2_^−^ (*m*/*z* 32), and C_5_H_5_^−^ (*m*/*z* 65), where O_2_^−^ and C_2_H^−^ were detected for all available energies. Although the molecular oxygen anion has a low electron affinity (0.448 eV [21]) in comparison with C_2_H^−^ (2.97 eV [21]), it yields up to 10% of all fragments formed, while C_2_H^−^ yields between 8% and 11% of the total fragments. This aspect is interesting from a biological damage point of view due to the O_2_^−^ anion’s [22] high reactivity. Moreover, the yield of fragment anion C_2_H_2_^−^ (26 *m*/*z*) is concealed by the isobaric fragment BO^−^ (26 *m*/*z*); nevertheless, the data from Figure 3 clearly shows how those isobaric anions contribute as a function of the energy. The formation of rather low-intensity ions assigned to C_3_H_2_B^−^ and C_3_H_2_BO^−^ may result from the LUMOs to effective ring breaking through σ* MOs (see Figure 8).

Finally, from the collision-induced dissociation of CHBA, sixteen different anions have been assigned in Table 1 with some of them not being detected in PBA’s TOF mass scans, including C_3_H_4_^−^/CBOH^−^ (*m*/*z* 40), C_5_H_10_BO^−^/C_4_H_6_BO_2_^−^ (*m*/*z* 97), and C_6_H_11_O^−^ (*m*/*z* 99).

## 3. Experimental and Theoretical Methods

### 3.1. Experimental Methods

The crossed molecular beam setup and experimental techniques used for the present study have been described in detail elsewhere [10,12], so these will not be explained in detail here. Briefly, a target molecular beam crosses a primary beam of neutral potassium (K) atoms at a given kinetic energy and the negative ions formed by electron transfer are time-of-flight (TOF) mass analysed using a commercial reflectron TOF mass spectrometer (KORE R-500-6). The experimental setup comprises the potassium chamber and the collision chamber, which are separated by a manual gate valve in order to guarantee differential pumping in both chambers. In the first chamber, the K beam is obtained from an accelerated K^+^ beam produced in a commercial source (Heatwave lab, Watsonville, CA, USA), which passes through an oven, where it resonantly charge-exchanges with thermal K to finally acquire a beam of hyperthermal K at a given kinetic energy. Residual ions are removed before the K beam enters the collision chamber by electrostatic deflecting plates outside the oven. Before and after the collision events, the hyperthermal K beam intensity is monitored using a Langmuir–Taylor detector. The effusive beam of target molecules from an oven source is admitted into a vacuum through a 1 mm diameter capillary, where it crosses with the K beam. Negative ions resulting from the collision between the neutral potassium beam and the molecular target are extracted by applying a ~380 V/cm electrostatic field pulse into the TOF spectrometer to finally reach a dual microchannel-plate (MCP) detector. The anion yields are obtained by normalizing the mass spectra to the acquisition time and the K beam intensity. The typical background pressure in the collision chamber is 6 × 10^−5^ Pa and the working pressure is ≈ 4 × 10^−4^ Pa. From the obtained mass spectra (resolution m/Δm ≈ 700), the background signal is subtracted (without the sample) and the mass calibration is performed on the basis of the well-known nitromethane negative ions formed upon potassium collision [23].

The phenyl boronic acid and the cyclohexyl boronic acid samples were purchased from Sigma-Aldrich with a minimum purity of 99% and used as delivered. In both cases, the samples were moderately heated up to 423 K through a temperature PID (proportional-integral-derivate controller) unit. In order to test for any thermal decomposition products within the target beam, mass spectra were recorded at different temperatures and no differences in the relative peak intensities as a function of temperature were observed.

### 3.2. Theoretical Method

The mechanism and quantitative characteristics of charge-exchange processes in K + boronic acid interaction requires a model which is simple enough to be computationally feasible, but which includes sufficient details of the problem to reproduce the essential features of the process. Therefore, based on our previous works [24,25,26], we adapted a simple framework of this polyatomic system of interaction between the potassium atom and the PBA molecule (K+PBA) constructed in the frame of one-dimensional reaction coordinate approximation. In this case, a projectile K approaches the corresponding target molecule through a single straight-line trajectory. With such an assumption, the collision can be represented as the evolution of the polyatomic K–PBA complex which, in a first approximation, can be treated as a pseudo-diatomic system. The reaction coordinates thus correspond to the distance R between the PBA molecule and the colliding potassium atom. Such a framework has already been applied to several cases, in particular, those of biological interest, in which an ion/atom collides with a molecular target composed of an aromatic ring [27,28]. Moreover, as previously pointed out for such types of collisions with potassium ions/atoms [12,29] in the considered range of energies, the collision time is much shorter (around 10^−14^ s) than typical vibration and rotation times. Taking this into account, since such an approach does not consider the degrees of freedom of the complex and the internal motions of the molecule, the geometry of the PBA target can be considered as frozen during the collision time.

The geometry of the PBA molecule consists a phenyl ring, as well as the B(OH)_2_ group. The anisotropy of the charge transfer process can be considered for different orientations of the projectile K atom toward this ring. The evolution of the system and the energies of the different electronic states can be calculated along any reaction coordinate R for different angles θ. In particular, we chose a perpendicular approach (θ = 90°) in which the potassium atom points along the side of the axis at the carbon atom directly connected to the B(OH)_2_ group. The molecular structure and the model of the collision are illustrated in Figure 7. First, the geometry of the ground ^1^A’ singlet state of the PBA molecule in the *C_s_* symmetry was optimized by means of density-functional-theory (DFT) calculations using the Becke–Lee–Yang–Parr density functional [30,31] (B3LYP) with the balanced polarized triple-zeta def2-TZVP basis set [32], which has been shown to be computationally efficient and provides accurate structures and transition energies. All calculations were performed by means of the ORCA 3.0.3 [33] and MOLPRO 2012.1 suite of ab initio programs [34,35].

The calculations were performed in Cartesian coordinates with no symmetry, while the PBA target optimized in *C_s_* symmetry was considered frozen during the collision process. The ECP18sdf core-electron pseudopotential with the associated basis set was chosen [36] for the potassium atom. The natural molecular orbitals for K + PBA were computed at a state-averaged Complete Active Space Self Consistent Field (CASSCF) [37] level of theory for the reaction coordinate K-C-B(OH)_2_ at R = 10 Å. The active space takes into account important orbitals of the PBA molecule and the potassium atom, and we considered one with CAS (5,10), which includes the five electrons distributed over valence and virtual orbitals. All electrons from B, C, and O atoms were included in the calculations and their 1 s orbitals were treated as core-frozen. The highest occupied molecular orbital (HOMO, 5a”) and the second highest occupied molecular orbital (HOMO–1, 4a”) of PBA in the neutral ground state have a π character on (C5-C6/C3-C2 and C4/C1-B) and (C5-C6-C1/C4-C3-C2) positions (see Figure 8), whilst HOMO–2 has a quite delocalized π character over the phenyl ring. The lowest unoccupied molecular orbital (LUMO) mainly displays π* antibonding and it is localized on the ring C–C bonds. The main LUMOs of PAB in the presence of potassium are shown in Figure 8, where LUMO+1, LUMO+2, and LUMO+3 have a π* character, while LUMO+4 and LUMO+5 mainly exhibit σ* antibonding along (C4-C5) and (C6-C1/C3-C2) bonds. Note that in similar collisions involving a pyrimidine molecule, the polarization induced by the potassium atom close to the target molecule shifts by ∼1.5–2.0 eV for the π* orbitals and by ∼2.0 eV for the σ* orbitals [12].

## 4. Conclusions

The present work provides, for the first time, an experimental and theoretical investigation of the dissociation pattern upon the electron transfer of two different boronic acid compounds: phenyl boronic acid (PBA) and cyclohexyl boronic acid (CHBA). A comprehensive analysis of the fragmentation pattern of both molecules as a function of energy has been presented. In the case of PBA, the results showed a very rich fragmentation pattern that led to the formation of several different boron-containing anions, particularly boron oxides, BO^−^ and BO_2_^−^, which were supported by ab initio theoretical calculations. In addition, it was shown that the main dissociation pathway at available energies below 100 eV leads to the formation of BO^−^, while for higher energies, the predominance is for BO_2_^−^. Moreover, it is worth noting that collision-induced dissociation is selective in terms of yielding abundant boron oxides, BO^−^ and BO_2_, which can be easily achieved by just tuning the proper collision energy. This may be of relevance to bio-signaling, resulting in an additional advantage of the use of boronic acids in cancer therapy. From a biological damage point of view, the formation of boron oxides also yields radical species such as H^●^ and OH^●^, which can be of damaging to either tumor cells or nearby healthy tissues.

The dissociation pattern of CNBA was investigated and compared with PBA, noting the differences stemming from the system dynamics and the promotion of different dissociation channels. However, further complementary studies are needed, in particular, comprehensive DEA experiments of both molecules, as well as ab initio calculations of the lowest unoccupied molecular orbitals (LUMOs) of cyclohexylboronic acid in the presence of a potassium atom.

Finally, we consider that the results obtained through this study further our knowledge on the dissociation channels involved upon electron transfer in two important boronic acids and can thus contribute to improving the performance and role of such chemical compounds in medical chemistry and biomedical engineering.

## Figures and Tables

**Figure 1 ijms-20-05578-f001:**
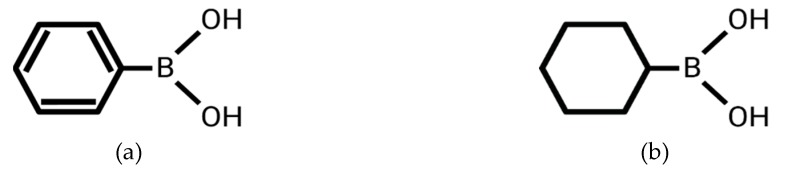
A schematic representation of the molecular structure of (**a**) phenyl boronic acid (PBA) and (**b**) cyclohexyl boronic acid (CHBA).

**Figure 2 ijms-20-05578-f002:**
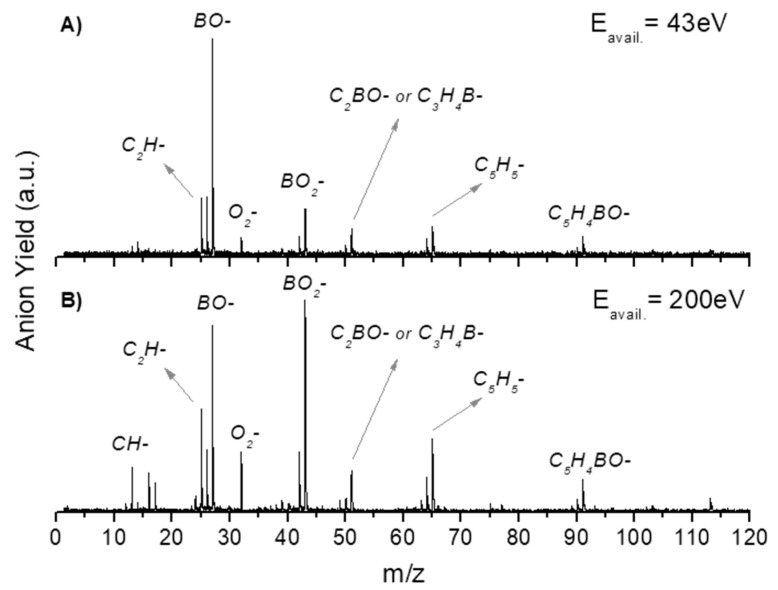
Time-of-flight negative ions mass spectra in neutral potassium collisions with phenylboronic acid at a 70 eV (**A**) and 300 eV (**B**) collision energy (43 eV and 200 eV available energy). The most intense fragments are assigned.

**Figure 3 ijms-20-05578-f003:**
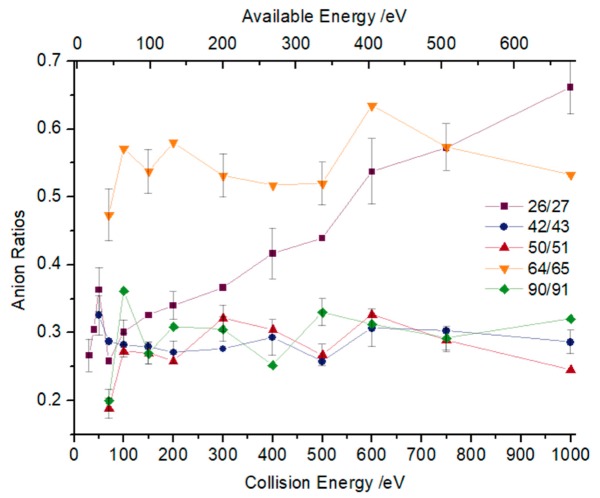
Ratio between the yield of relevant anions that differ by one *m*/*z* as a function of the energy. Error bars related to the experimental uncertainty associated with the ion yields have been added to a few data points in order to avoid congestion of the figure. The lines are just to guide the eye.

**Figure 4 ijms-20-05578-f004:**
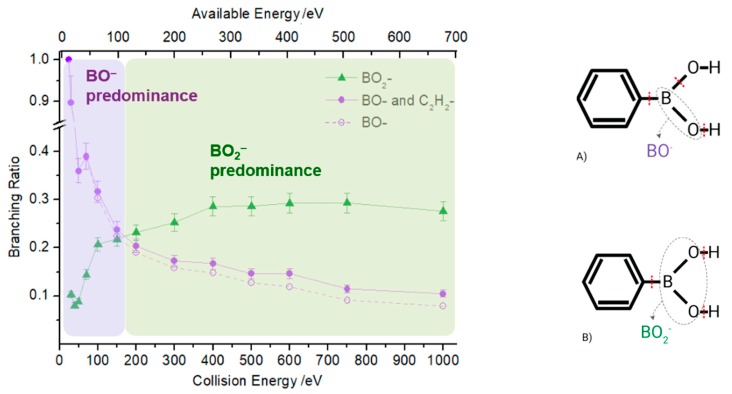
Left: PBA branching ratio of BO^−^ and BO_2_^−^ ions as a function of the collision energy. Green lines with full triangles represent BO_2_^−^, while purple lines with full circles indicate BO^−^ with the contribution of the isobaric fragment C_2_H_2_^−^. The dashed line represents the BO^−^ branching ratio without the contribution of C_2_H_2_^−^ and was obtained based on the natural abundance of heavier isotopic species (^11^BO^−^), not contaminated by any other fragment. Right: schematics of bond rupture leading to BO^−^ (**A**) and BO_2_^−^ (**B**) formation.

**Figure 5 ijms-20-05578-f005:**
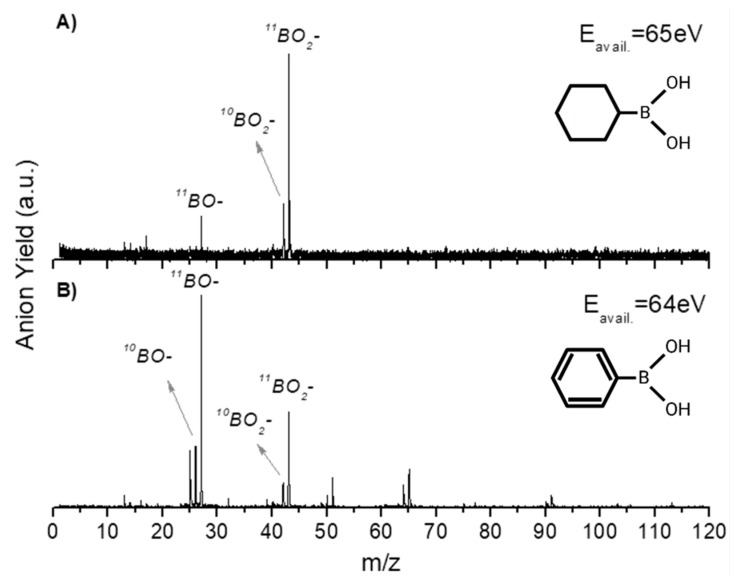
Comparison of time-of-flight negative ions’ mass spectra in neutral potassium collisions with CHBA (**A**) and PBA (**B**) at a 100 eV collision energy (~65 eV available energy).

**Figure 6 ijms-20-05578-f006:**
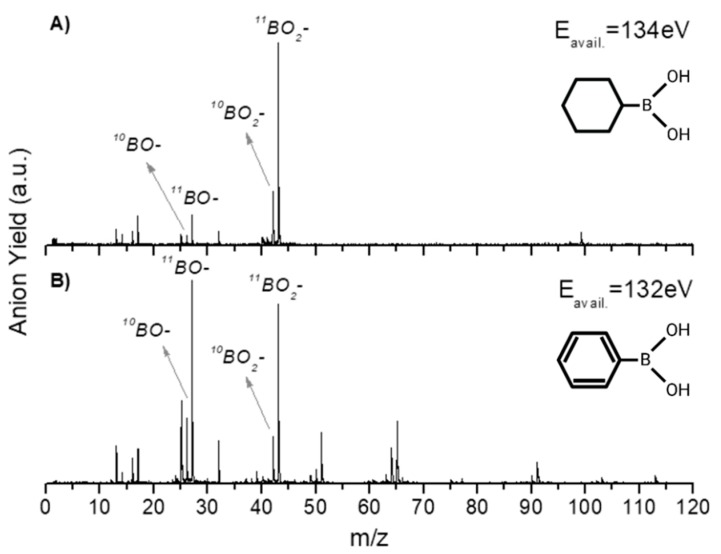
Comparison of time-of-flight negative ions’ mass spectra in neutral potassium collisions with CHBA (**A**) and PBA (**B**) at a 200 eV collision energy (~132 eV available energy).

**Figure 7 ijms-20-05578-f007:**
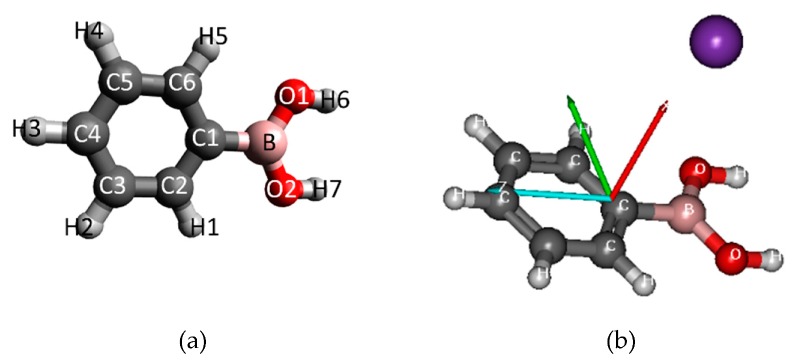
(**a**) Molecular structure of PBA, including atom numbering (**b**) and orientation of the potassium (K) + PBA collisional system.

**Figure 8 ijms-20-05578-f008:**
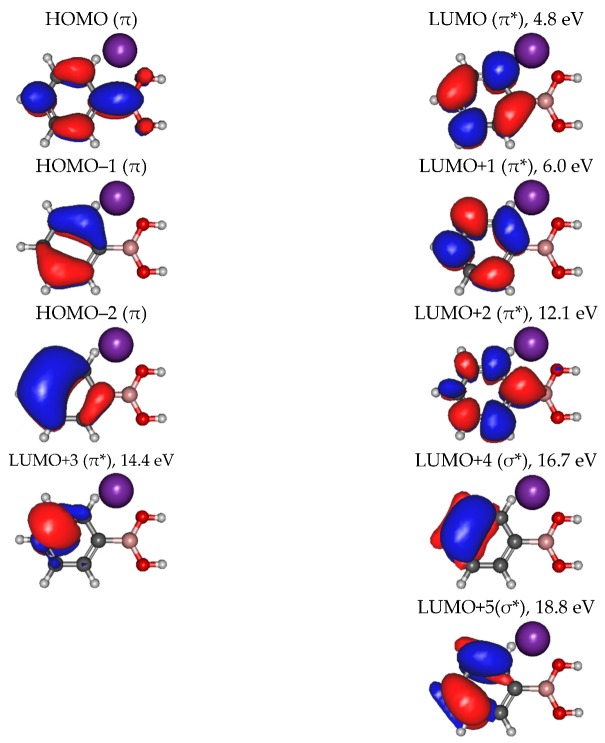
Calculated highest occupied molecular orbital (HOMO) and lowest unoccupied molecular orbitals (LUMOs) for phenylboronic acid (PBA). Orbitals were calculated at the CASSCF (5,10) level of theory in the presence of a potassium atom (in purple color) at a distance of 10 Å.

**Table 1 ijms-20-05578-t001:** Assignment of negative ions formed in neutral potassium collisions with phenyl boronic acid (PBA) and cyclohexyl boronic acid (CHBA).

Mass (a. m. u.)	PBA	CHBA
103	C_6_H_4_BO^−^	
99		C_6_H_11_O^−^
97		C_5_H_10_BO^−^/C_4_H_6_BO_2_^−^
91	C_5_H_4_^11^BO^−^	
90	C_5_H_4_^10^BO^−^	
89	C_5_H_2_^11^BO^−^/C_6_H_6_^11^B^−^	
77	C_6_H_5_^−^	
75	C_6_H_3_^−^	
66	C_3_H_3_^11^BO^−^	
65	C_5_H_5_^−^	
64	C_5_H_4_^−^/C_3_H_2_^10^BO^−^/C_3_H^11^BO^−^/C_4_H_5_^11^B^−^	
63	C_5_H_3_−/C_3_H^10^BO^−^/C_4_H_5_^10^B^−^	
51	C_2_^11^BO^−^/C_3_H_4_^11^B^−^	
50	C_2_^10^BO^−^/C_3_H_4_^10^B^−^	
49	C_3_H_2_^11^B^−^	
46	^11^BO_2_H_3_^−^	
43	^11^BO_2_^−^	^11^BO_2_^−^
42	^10^BO_2_^−^	^10^BO_2_^−^
41		CH_2_BO^−^/C_3_H_5_^−^
40	C^11^BOH^−^/C_3_H_4_^−^	C^11^BOH^−^/C_3_H_4_^−^
39	C^11^BO^−^/C_3_H_3_^−^	C^11^BO^−^/C_3_H_3_^−^
38	C_2_H_3_^11^B^−^/C_3_H_2_^−^	
37	C_2_H_2_^11^B^−^/C_3_H^−^	
32	O_2_^−^	O_2_^−^
30	CH_2_O^−^	
27	^11^BO^−^	^11^BO^−^
26	^10^BO^−^/C_2_H_2_^−^	^10^BO^−^
25	C_2_H^−^	C_2_H^−^
24	CH^11^B^−^/C_2_^−^	
17	OH^−^	OH^−^
16	O^−^	O^−^
14	CH_2_^−^	CH_2_^−^
13	CH^−^	CH^−^
12	C^−^	C^−^

**Table 2 ijms-20-05578-t002:** Electron affinity of neutral species [19].

Neutral Species	Electron Affinity (eV)
BO	2.510 ± 0.015
BO_2_	4.460 ± 0.030
C_2_H^•^	2.9689 ± 0.0011
C_2_H_2_	0.480 ± 0.010
C_5_H_5_^•^	1.786 ± 0.020
C_5_H_4_	1.750 ± 0.047
C_6_H_5_^•^	1.0960 ± 0.0060
O	1.4610 ± 0.0010
OH•	1.829 ± 0.010
O_2_	0.4480 ± 0.0060
C_6_H_11_^•^	−0.24 ± 0.11

Note that ^•^ refers radical species.

**Table 3 ijms-20-05578-t003:** Dissociation energies of the most relevant molecular bonds [20].

Bond	Dissociation Energy (eV)
C–B	4.640 ± 0.030
O–B	8.38
O–H	4.460 ± 0.003
C–C	6.408 ± 0.160

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
