# Peer review of "The Role of Electron Transfer in the Fragmentation of Phenyl and Cyclohexyl Boronic Acids"

_ijms, 2019, doi:10.3390/ijms20225578_

Round 1
Reviewer 1 Report
Comments on the Manuscript Number: ijms-636685
Title: "The role of electron-transfer in the fragmentation of phenyl and cyclohexyl boronic acids”
Authors: Ana Isabel Lozano, Beatriz Pamplona, Tymon Kilich, Marta Łabuda, Mónica Mendes, João Pereira-da-Silva, Gustavo García, Pedro Góis , Filipe Ferreira da Silva and Paulo Limão Vieira
This paper presents a mass spectroscopy study of two molecules (derivatives of boronic acid) paying special attention to formation of negative species. They bombard the molecules with K atoms of different energies and then analyze the different fragments found in the experiments. First-principles calculations are used to devise the most relevant molecular orbitals of the molecules to help the discussion of the experiments.
The paper is ordered and the results clearly stated. In spite of the moderate interest of the main conclusion for a general audience I have no serious comments about the scientific reliability of this work. As a minor comment, I suggest to give some more details of the experimental methods used. It is true that there are previous references in this respect but it would be helpful for the reader to enumerate and make a brief description of the experimental techniques. Regarding the details of the calculations it is not clear how the orbitals of Fig. 3 have been calculated. Are which distance is placed the K atoms and why? It has been studied the variations of the molecular orbitals with respect to the molecule-K distance?
Author Response
Reviewer #1
As a minor comment, I suggest to give some more details of the experimental methods used
Authors’ reply: changed accordingly.
It is true that there are previous references in this respect but it would be helpful for the reader to enumerate and make a brief description of the experimental techniques
Authors’ reply: short sentences added as per in the reviewer comment.
Regarding the details of the calculations it is not clear how the orbitals of Fig. 3 have been calculated
Authors’ reply: authors still believe that the description in section 2.2 and references therein are enough and so any other description would make this section rather extensive. In any case, Figure 3 legend has been changed in order to give further details on the calculation as requested.
Are which distance is placed the K atoms and why? It has been studied the variations of the molecular orbitals with respect to the molecule-K distance?
Authors’ reply: The distance between the K atom and molecule is taken to be 10 Å. This distance was assumed in the model (described more thoroughly in references 15-17) and has shown to reproduce key features of electron transfer process in such systems. Particularly, it allows for an estimation, which orbitals are taking a part in this process and from which orbital an electron is excited. Indeed, apart from the chosen R=10 Å between K and PBA, the variations of molecular orbitals were studied for various distances between 15 and 5 Å and the results have shown that there are essentially no qualitative changes in the shape of the orbitals.
Reviewer 2 Report
The manuscript "The role of electron-transfer in the fragmentation of phenyl and cyclohexyl boronic acids" by Ana Isabel Lozano and co-workers reports on molecule fragmentation in slow collisions of K atoms with PBA and CHBA molecules. The reported experimental work is of good quality, the modelling of the excited molecular states looks reasonable and the interpretation of the results are sound. The article is well suited for IJMS. However, there several points which need to be resolved before the manuscript could be published.
/abstract/ I don't like statements like "for the first time" and would urge the authors to keep the abstract in a more scientific fashion without such advertisements.
/lines 77-81/ This is a pure space filler and should be removed.
/l.85/ delete from "thereby" to and of sentence
/l. 121/ the given collision time of 1E-16 s is much to short. Later the authors give a value of 200 fs for their projectile at the chosen energy. A value of 1E-14 s seems more appropriate.
/l. 177/ Here and at many more places the text "Error! Reference source not found" appears. Please insert correct references.
/l. 202/ change "m/z 64 and 65" to m/z=64 and m/z=65
/l. 204/ dito.
/l. 206-8/ "such anion contribution becomes relevant where this is mostly related to opening of another dissociation channel yielding an isobaric contribution of 10 BO ‒ / 11 BO ‒ which is assigned to C 2 H 2–" rewrite this sentence to make more clear that you are talking about the superposition of signals from two masses. The way the sentence is written is quite complicated to understand.
/l. 219-33/ This paragraph is rather confusing. Authors mention 12.8 eV available energy, but dicuss promotion to LUMO+3 and LUMO+4 states with higher excitation energies? Maybe this (and the next paragraph) could be shortened because the more important discussion mentioning all bonds which need to be broken starts at line 253.
/l. 256-60/ Why is this paragraph in smaller font?
/eq. 3/ Shouldn't his be added to eq. 1, similar to eq. 2a and 2b?
/l. 362/ The fragments C3H4- and CBOH- have mass 40, not 41. According to table 1 CH2BO- and C3H5- have m/z=41.
Author Response
Reviewer #2
/abstract/ I don't like statements like "for the first time" and would urge the authors to keep the abstract in a more scientific fashion without such advertisements.
Authors’ reply: done.
/lines 77-81/ This is a pure space filler and should be removed.
Authors’ reply: removed as requested.
/l.85/ delete from "thereby" to and of sentence
Authors’ reply: done.
/l. 121/ the given collision time of 1E-16 s is much too short. Later the authors give a value of 200 fs for their projectile at the chosen energy. A value of 1E-14 s seems more appropriate.
Authors’ reply: this has been changed accordingly.
/l. 177/ Here and at many more places the text "Error! Reference source not found" appears. Please insert correct references.
Authors’ reply: this was probably a result of the conversion mechanism in the journal’s platform when uploading the manuscript, which authors did not observe at all in the original word file.
/l. 202/ change "m/z 64 and 65" to m/z=64 and m/z=65
Authors’ reply: changed.
/l. 204/ dito.
Authors’ reply: changed.
/l. 206-8/ "such anion contribution becomes relevant where this is mostly related to opening of another dissociation channel yielding an isobaric contribution of 10 BO ‒ / 11 BO ‒ which is assigned to C 2 H 2–" rewrite this sentence to make more clear that you are talking about the superposition of signals from two masses. The way the sentence is written is quite complicated to understand.
Authors’ reply: done.
/l. 219-33/ This paragraph is rather confusing. Authors mention 12.8 eV available energy, but dicuss promotion to LUMO+3 and LUMO+4 states with higher excitation energies? Maybe this (and the next paragraph) could be shortened because the more important discussion mentioning all bonds which need to be broken starts at line 253.
Authors’ reply: although we do understand the reviewer comment, authors still consider that this paragraph is important as an explanation about why at low energies the BO- yield is found to be higher than of BO2-
/l. 256-60/ Why is this paragraph in smaller font?
Authors’ reply: we are very sorry but in the original word file we could not spot such style issue.
/eq. 3/ Shouldn't his be added to eq. 1, similar to eq. 2a and 2b?
Authors’ reply: changed.
/l. 362/ The fragments C3H4- and CBOH- have mass 40, not 41. According to table 1 CH2BO- and C3H5- have m/z=41.
Authors’ reply: changed.